# Clinical Phenotypes of COVID-19 Associated Mucormycosis (CAM): A Comprehensive Review

**DOI:** 10.3390/diagnostics12123092

**Published:** 2022-12-08

**Authors:** Maria Panagiota Almyroudi, Karolina Akinosoglou, Jordi Rello, Stijn Blot, George Dimopoulos

**Affiliations:** 1Department of Emergency Medicine, University Hospital Attikon, Medical School, National and Kapodistrian University of Athens, 12462 Athens, Greece; 2Department of Internal Medicine and Infectious Diseases, University General Hospital of Patras, School of Medicine University of Patras, 26504 Rio, Greece; 3Vall d’Hebron Institute of Research, Barcelona, Spain & Clinical Research, CHU Nîmes, 30900 Nîmes, France; 4Department of Internal Medicine and Pediatrics, Ghent University, 9000 Ghent, Belgium; 5UQ Centre for Clinical Research, Faculty of Medicine, The University of Queensland, Brisbane 4029, Australia; 63rd Department of Critical Care, EVGENIDIO Hospital, Medical School, National and Kapodistrian University of Athens, 11528 Athens, Greece

**Keywords:** mucorales, invasive fungal infections, SARS-CoV-2, diabetes mellitus, rhino-orbito-cerebral mucormycosis

## Abstract

A mucormycosis surge was reported during the COVID-19 pandemic in India. A literature search until 14 July 2022, with the aim of updating COVID-19-associated mucormycosis (CAM), identified 663 studies and 88 met inclusion criteria (8727 patients). India reported 8388 patients, Egypt 208 and Europe 40. Rhino-orbito-cerebral mucormycosis (ROCM) was identified among 8082 (98.3%) patients, followed by 98 (1.2%) with pulmonary. In India, 82.6% of patients had diabetes mellitus, with 82% receiving corticosteroids. In Europe, 75% presented pulmonary CAM, 32.5% had diabetes and 40% were immunocompromised. CAM was identified at a median of 17.4 days (IQR 7.5 days) post COVID-19 diagnosis, and PCR was performed in five studies. Rhino-orbital invasion is clinically obvious, while cerebral involvement presents with cavernous sinus thrombosis, meningitis and cerebrovascular disease. Symptoms of pulmonary CAM usually overlap with severe COVID-19 pneumonia. High-dose liposomal Amphotericin B (and early surgical debridement in ROCM) are the mainstay of therapy. The median mortality rate was estimated to be 21.4% (IQR 31.9%), increased by the presence of pulmonary (80% (IQR 50%) or cerebral involvement (50% (IQR 63.9%). In summary, different CAM clinical phenotypes need to be distinguished, influenced by geographical presentation. Opportunities exist for diagnosis and therapy optimization, based on earlier high-dose antifungal therapy, early source control, strict glycemic control and restriction of steroids to COVID-19 patients with oxygen requirements.

## 1. Introduction

Mucormycosis is a rare infection caused the members of the order *Mucorales*. Its prevalence ranges from 0.005 to 1.7 per million people worldwide, while in India, it reaches 14 cases per 100,000 inhabitants [1,2]. During the COVID-19 pandemic, a surge in mucormycosis cases has been observed, especially in India, where the Government of India portal reported 47,508 cases from 5 May 2021 to 3 August 2021 [3]. Characteristically, Samir Joshi et al. reported 160 cases of COVID-19-associated mucormycosis (CAM) from April to May 2021 in the Ear, Nose, Throat Department of BJGMC-SGH hospital in India, compared with 3–8 cases of mucormycosis detected each year from 2016 to 2020 [4].

Invasive fungal infections may complicate COVID-19, as immunological alterations, intense inflammatory response and lung damage favor fungal growth [5]. Aspergillosis is most often reported, but mucormycosis has also emerged and been associated with COVID-19. The most common fungi isolated are *Mucor* and *Rhizopus*, followed by *Cunninghamella* sp., *Saksenaea* sp., *Lichtheimia* sp., *Apophysomyces* sp., *Rhizomucor* sp. and *Cokeromyces* sp. [5]. They are ubiquitous in nature and the spores are transmitted mainly by inhalation, leading to sinus or lung infection, by ingestion or by direct inoculation following trauma. They cause invasive disease in vulnerable patients with predisposing medical conditions and risk factors including immunosuppression, diabetes mellitus (DM), corticosteroid treatment, hematologic malignancies, hematopoietic cell transplantation, solid organ transplantation and iron overload [6,7]. Different types of mucormycosis are recognized, with rhino-orbito-cerebral mucormycosis (ROCM) and the pulmonary form being the most common clinical presentations [6].

The fungus usually first infects the nasal mucosa and the palate and subsequently spreads to paranasal sinuses and to the retro-orbital space through the ethmoid sinus [8]. It spreads through direct regional extension and vascular and perineural invasion, while angioinvasion results in thrombi formation, tissue infarction and necrosis. Immune dysregulation with impaired phagocytosis, endothelial dysfunction, hyperglycemia, hypoxia, acidosis and hyperferritinemia, seen with COVID-19, create favorable conditions for fungal growth [5]. *Mucorales* spores escape phagocytosis and transform into hyphae, enabling tissue invasion [5]. Neutrophil dysfunction, in the context of COVID-19, DM and steroid use, plays an important role in the pathogenesis of CAM. As macrophage function and polymorphonuclear chemotaxis is impaired in diabetics, spores escape phagocytosis, germinate, enlarge and convert to hyphae which invade the tissues and vessels [2]. Additionally, in an acidic environment, as in diabetic ketoacidosis, the binding capacity of transferrin is reduced and free iron is released into the circulation. Iron is then absorbed in a solubilized form by *Mucorales* spores. Elevated ferritin levels are commonly found in COVID-19 patients, reflecting high iron levels, which act as a substrate for fungal overgrowth [2].

The aim of this review is to present the current data in the literature concerning the incidence, risk factors, pathophysiology, diagnosis, outcome and treatment of CAM.

## 2. Methods

The literature was explored using the search string “((COVID *) OR (SARS-CoV2) OR (SARS-CoV-2)) AND (mucor *)” in Pubmed (EMBASE). Observational trials, interventional trials and case series (the latter consisting of ≥5 CAM patients) until 14 July 2022 were considered for inclusion. Only studies written in English, with available full article and using the diagnostic criteria for mucormycosis proposed by the European Confederation of Medical Mycology in cooperation with the Mycoses Study Group Education and Research Consortium [6] were included. In addition, studies including patients with invasive fungal infections in general, and not mucormycosis specifically, and studies including mucormycosis cases not associated with COVID-19 and not separately with CAM cases were excluded. Sixteen studies were excluded due to unavailable–insufficient information [9,10,11,12,13,14,15,16,17,18,19,20,21,22,23,24]. Four studies were excluded as they included patients with non-microbiologically confirmed SARS-CoV-2 infection [25,26,27,28]. Figure 1 reports a PRISMA flow-chart detailing the selected studies and exclusion criteria.

The quality of the included articles was assessed by means of the Quality Assessment Tool for Case Series Studies by the National Heart, Lung and Blood Institute (NHLBI). Two authors independently screened the literature search results and executed data extraction (MPA, GD). In each study, data regarding the type of the study, the country of origin, the setting of actualization, the number of CAM patients, the reported CAM incidence, the interval between COVID-19 and CAM diagnosis, the type of infection, the comorbidities and risk factors for CAM, the performance of PCR and the identification of species and the outcome were sought and registered when they were mentioned. The median mortality rate and median interval between COVID-19 diagnosis and CAM diagnosis were calculated according to the formula p = (n + 1)/2 where n is the size of the data set and p is the position of the median value. The PRISMA 2020 checklist is detailed in Appendix B.

## 3. Incidence of CAM

The literature search revealed 662 studies, of which, ultimately, 88 studies were included in the review. Overall, only 13/88 studies were prospective. According to the Quality Assessment Tool for Case Series Studies by the National Heart, Lung and Blood Institute (NHLBI), 32 studies were rated as good quality studies, 42 as fair quality studies and 14 as poor quality studies (Table 1).

A total of 8727 cases of CAM are reported in the 88 qualified studies (Table 1), the majority of them from India (n = 8388, 96.1%), 208 patients (2.4%) were from Egypt and 40 patients (0.46%) from Europe (France and Germany) (Figure 2). CAM cases have also been reported in USA [41], Pakistan [70], Iran [84], Oman [113], Syria [115], Iraq [116], Latin America [117] and Turkey [7].

The incidence of CAM largely remains unknown, as 12 studies out of 88 have mentioned CAM incidence (Table 2). In India, CAM incidence (September 2020–June 2021) ranged from 0.27% to 3.36% among hospitalized and microbiologically confirmed COVID-19 patients and the rhino-orbital-cerebral form represented the majority of cases (Table 2) [32,44,61,80,82,102]. Among seven centers in India, the prevalence of CAM was estimated to be 0.27% in general wards (28/10,517 COVID-19 patients) and 1.6% (25/1579) in ICUs [102]. However, Casalini et al., reported an average incidence of 8.6% among hospitalized COVID-19 patients in a review including 12 observational studies and 3126 cases of CAM [118]. A higher incidence was estimated among kidney transplant COVID-19 patients in India in two studies (4.4% and 10.8%) [40,42] Notably, during a 6-month period of the pandemic, 61 kidney transplant patients were diagnosed with CAM, in contrast with only 11 patients who had been diagnosed with mucormycosis during the previous year [42].

Similar characteristics are observed in Egypt. Sahar Farghly Youssif et al. reported an incidence of CAM of 7.6% among 433 patients with confirmed COVID-19 infection [107]. ROCM was identified in 207/208 (99.5%) CAM patients with only 1/208 (0.5%) having lung involvement.

On the other hand, in Europe, the reported incidence is much lower, but still, COVID-19 patients seem to be more affected, and interestingly, the pulmonary form is the most prevalent clinical presentation (Table 2). In France, among 473,353 COVID-19 patients hospitalized from March 2020 to June 2021, only 17 cases were considered as CAM nationwide (0.0036%) [57]. In two German tertiary centers, the CAM prevalence from January 2020 to June 2021 was 0.67% and 0.58%, respectively, while it was much lower in non-COVID-19 patients (0.0047% and 0.001%, respectively). In the same centers, in the ICUs, the prevalence of CAM was 1.47% and 1.78%, while in non-COVID-19 patients the prevalence of mucormycosis was 0.015% and 0.005% [76].

In the ICU, the prevalence of mucormycosis was estimated at 0.3–0.8% of COVID-19 admissions, based on data from four centers in France, Germany, Mexico and Turkey [7], while in a national, observational cohort study in 18 ICUs across France, among 509 COVID-19 mechanically ventilated patients who were systematically screened for respiratory fungal infections, 6 (1%) were diagnosed with invasive mucormycosis [119]. Analogously, in another German study, among 100 COVID-19 ICU patients whose respiratory specimens were checked routinely for Mucorales via culture and PCR, only one PCR in the bronchoalveolar lavage (BAL) came out positive for Mucorales (1%) [120]. However, a cluster of cases was detected in a French ICU, during a two month period, which was possibly linked to construction work that was undertaken near the ICU [62].

## 4. Risk Factors

Generally, 82.4% (6824/8279) of CAM patients were either diabetics or developed hyperglycemia during COVID-19 illness. In India, Egypt and Europe, 82.6%, 84.6% and 32.5% of patients had DM, respectively (Table 2). Poorly controlled DM is the strongest risk factor for mucormycosis described in the literature. In a case control study, the level of hyperglycemia was associated with the risk of CAM development, with the highest risk reported for patients with blood glucose >400 mg/dL [36]. Among 26 patients with CAM, the mean HbA1c was 9.3% (8–10.7%) [33], while diabetic ketoacidosis was noticed in 40% (27/67) of patients with CAM [32]. Except for preexisting DM, hyperglycemia that develops during COVID-19 illness, induced by the virus infection (viral-mediated islet cell destruction) or attributed to steroid treatment, contributes, respectively, to CAM emergence.

Corticosteroids have been established as the standard therapy for severe COVID-19, which increases the risk of secondary infections [121]. Overall, 81.6% (5777/7058) of patients with CAM had received steroids, 82% of Indian cases, 81.6% of Egyptian and 82.5% of European (Table 2). In many cases, steroid administration was considered as inappropriate, as they were given even in mild COVID-19 disease [33]. Finally, in a large Indian study, among 2826 patients with CAM, only 2% (n = 47) were neither diabetic, nor had they received corticosteroids [83], underlining the determinant role of these risk factors. Immunosuppression, hematological malignancies and solid organ or bone marrow transplantation are also well-known risk factors for invasive fungal infections. Forty percent of CAM patients in Europe were immunosuppressed, while 7/73 (9.6%) of patients with COVID-19-associated pulmonary mucormycosis had hematological malignancy. Danion et al. [57] described 17 cases of CAM in France, where, in contrast to Indian reports, fewer patients were diabetics (47%) and a higher proportion had hematological malignancies (41%) [57]. Other comorbidities identified in the literature are hypertension, chronic kidney disease, liver disease and ischemic heart disease (Appendix A).

Ultimately, whether COVID-19 independently predisposes patients to mucormycosis infection needs to be studied further. In a prospective cohort study, among 540 proven cases of mucormycosis from March to May 2021, 89.4% of patients had a history of previous COVID-19 infection [31]. Two hundred and eighty-eight/2801 (10.3%) of CAM patients had been submitted to invasive or non-invasive mechanical ventilation during hospitalization for COVID-19 according to data contained in 32 studies (Table 2). However, the severity of COVID-19 pneumonia does not seem to determine the development of COVID-19-associated ROCM. In a prospective observational study from India, 7.9% of 101 patients with post-COVID-19 ROCM were asymptomatic, 36% had mild disease, 40% had moderate disease and only 15.8% had severe disease [58]. Vare et al. emphasized that 22% of cases did not receive any supplemental oxygen [32], while according to CT criteria, the average severity of COVID-19 pneumonia complicated by ROCM was described as moderate [49]. The emergence of ROCM in non-severe COVID-19 is either associated with the determinant role of DM in the pathogenesis of CAM or may be related to the early death of critically ill patients before symptoms of mucormycosis develop. However, pulmonary mucormycosis is more often described in ICU patients, reflecting a possible association with severe COVID-19 pneumonia (Appendix A).

In multivariable analysis, in a case control study, risk factors that were independently associated with CAM were diabetes, glucose levels >200 mg/dL during the course of COVID-19, steroid use, mild and moderate (vs. severe) COVID-19 and repeated swab tests [36]. Additionally, in two studies, ferritin levels were significantly higher in patients with CAM compared to COVID-19 patients without mucormycosis [36,40] Zinc consumption is another risk factor discussed in the literature. Zinc was widely used as a nutrient supplement during the second wave of the pandemic in India and was significantly associated with CAM in a case–control study [60]. However, the relationship between zinc exposure and CAM remains controversial, as another study revealed opposite results, with zinc supplementation being more frequently used among COVID-19 patients without mucormycosis (79.9% vs. 53.8%, *p* < 0.001) [36].

Especially in India, the high prevalence of CAM is attributed to regional environmental factors, especially climatological conditions (hot and humid), to high incidence of uncontrolled DM and possibly to poor healthcare system conditions, as transmission of fungal spores through water used for oxygen humidifiers is speculated [8,118]. In a multi-centre study, including 11 hospitals in India, Mucorales contamination of 11.1% of air-conditioning vents was found, mainly with *Rhizopus* spp. [122]. CAM was also associated with prolonged use of cloth masks (4–6 h, *p* = 0.002; >6 h, *p* < 0.0001) and surgical masks (>6 h, *p* = 0.002) [36].

## 5. Clinical Presentation

CAM was diagnosed after a median of 17.4 days (Q1:14.4, Q3:21.8, IQR 7.5 days) post COVID-19 diagnosis (Appendix A) but simultaneous manifestation with acute COVID-19 is also reported. Mucormycosis may be associated with neuroinflammation of the acute phase or be integrated in the post-COVID-19 syndrome [123]. However, the long period that is mediated between COVID-19 positivity and CAM diagnosis may actually reflect a delay in diagnosis, that may be associated with a higher mortality [33].

Mucormycosis most commonly affects the head and neck region. ROCM is the commonest form globally and was also the most frequent form associated with COVID-19. ROCM was diagnosed in 8082/8218 (98.3%) CAM patients and pulmonary infection in 98/8218 (1.2%), of whom 30.6% were in Europe. (Table 2). Mucormycosis of the gastrointestinal tract was found in 5/8218 (0.06%) CAM patients, cutaneous in 11/8218 (0.13%), disseminated in 11/8218 (0.13%) and renal in 1/8218 (0.01%) (Table 2). In Europe 3/40 (7.5%) of CAM patients had ROCM, 30/40 (75%) had pulmonary mucormycosis, 4/40 (10%) mucormycosis of the gastrointestinal tract and 3/40 (7.5%) disseminated (Figure 3).

Clinical manifestations of head and neck mucormycosis include headache, loosening of teeth, black necrotic turbinate, facial pain, facial palsy, peri-orbital or facial swelling, skin induration and blackish discoloration [35]. Symptoms attributed to nasal and oral cavity invasion include epistaxis, bloody nasal discharge and palate destruction. Orbital extension may lead to destruction of the ophthalmic artery and optic nerves resulting in ptosis of the eyelid, proptosis, vision disturbances and blindness. In a large retrospective study from India, 519/2716 (19%) patients with CAM presented with vision loss [83]. Cavernous sinus involvement occurs due to extension from the orbit and manifests as diplopia and ophthalmoplegia [8].

Cerebral involvement was noted in 1400/7388 (18.9%) patients with COVID-19 associated ROCM, reported in 72 studies (Table 2). Cerebral involvement may manifest as cavernous sinus thrombosis, fungal abscess, meningitis and cerebrovascular disease [73]. Rahul Kulkarni et al. noted that 45/49 (91.8%) of patients with cerebral involvement presented with ischemic stroke, which concerned large artery infracts, followed by intracranial hemorrhage in 3/49 (6.1%) and sub-arachnoid hemorrhage in 1/49 (2.0%) [44].

Ninety-eight patients with pulmonary mucormycosis are described in the literature (57 in India, 30 in Europe, 6 in Pakistan, 3 in USA, 1 in Egypt and 1 in Mexico) (Appendix A). Ten studies contained data on ventilatory support, with invasive or non-invasive mechanical ventilation reported in 26/45 (57.8%) of patients with COVID-19-associated pulmonary mucormycosis. In 7 studies where CAPA was sought, 19/43 (44.2%) patients with pulmonary mucormycosis were found with positive microbiological testing for Aspergillus. Symptoms of pulmonary mucormycosis in non-ventilated patients include fever, dyspnea, cough, chest pain and hemoptysis [124]. Pruthi et al. reported five cases of pulmonary mucormycosis associated with COVID-19 that were complicated by pulmonary artery pseudoaneurysm [39]. In mechanically ventilated patients, identification of an agent of mucormycosis from respiratory specimens in combination with compatible radiographic findings support the diagnosis.

Symptoms of mucormycosis of the gastrointestinal tract are non-specific and consist of abdominal pain and distension, diarrhea and gastrointestinal bleed [124], while disseminated mucormycosis may affect any organ, but mainly the brain and lungs, and is a result of bloodstream invasion in severely immunocompromised patients [124].

## 6. Diagnosis

Early recognition of CAM is crucial, as delay in therapy is associated with higher mortality [42]. A high index of suspicion should be maintained when clinical symptoms and radiological features appear in a patient with predisposing factors. According to criteria proposed by the European Confederation of Medical Mycology and the Mycoses Study Group Education and Research Consortium [6], the diagnosis of mucormycosis is based on clinical and imaging characteristics and confirmed with direct microscopy, histopathologic analysis and culture of samples obtained with biopsy.

Diagnosis is challenging, as appropriate specimens are obtained through invasive procedures and specific stains are needed to identify Mucorales. Direct microscopy with potassium hydroxide (KOH) mount is usually used for the rapid diagnosis of mucormycosis, as results are delayed with culture and histopathology. Direct microscopy reveals wide, non-septate, ribbon-like hyaline hyphae, with irregular right-angled branching that are characteristic of Mucorales [125]. Ιnfarcts, angioinvasion and perineural invasion are usually present in the histological analysis. Preceding antifungal therapy may alter morphological characteristics of the fungus, while specimens’ processing must be carefully undertaken to keep hyphae intact [2]. Even when fungal hyphae are recognized in histopathological analysis, cultures may be negative in 50% of cases, due to the fragility of fungal hyphae [125]. Characteristically, among 2175 patients with CAM, direct microscopy with KOH/calcofluor white was performed in 89% (1931), and culture in 19% (432), of cases [83].

Molecular techniques are promising, as rapid detection is needed and cultures are time-consuming and may be false-negative [124]. However, results should be cautiously evaluated due to ubiquitous nature of Mucorales. PCR was performed in five studies in the literature (two French, two from India and one from Egypt) and concerned 174 patients, of which 31 were positive (Appendix A). In the French study [57], PCR was positive in 15/17 (88%) patients with CAM in serum (n = 14), BAL (n = 7), tissues (n = 3) and peritoneal fluid (n = 1) [57]. It is of interest that, in another French study [62], Mucorales was detected with PCR in respiratory samples of 10 COVID-19 patients, of which 80% simultaneously tested positive for Aspergillus. This cluster of cases was possibly attributed to environmental exposure, due to construction work near the hospital [62].

Few studies in the literature report on the species isolated, reflecting the difficulties encountered with culture-based identification and the infrequent use of PCR. Rhizopus sp. were the most common species isolated (Appendix A). In a study including 203 cases of mucormycosis with positive cultures during the second wave of the pandemic in India, Rhizopus oryzae, followed by R. microspores, were most frequently identified [126].

Mixed infections with Aspergillus and Candida are detected both in pulmonary and rhino-orbital-cerebral form. Eighteen studies in the literature (12 from India, 3 from Europe, 2 from Pakistan and 1 from Egypt) refer to Aspergillus possible co-infection, with Aspergillus being isolated in 89/863 (10.3%) CAM patients (Appendix A). Danion et al., reported 5 mixed fungal infections with Aspergillus in 17 (29%) CAM cases, of which 2 exhibited pulmonary involvement, 1 ROCM, 1 disseminated and 1 GI disease. All patients were mechanically ventilated and COVID-19-associated pulmonary aspergillosis (CAPA) was diagnosed at a median of 2 days before CAM. Four out of five patients with CAM and CAPA received L-Amphotericin B (one was diagnosed after death) and 5/5 died [57]. In Toulouse, France, eight cases of concomitant infection with Mucor and Aspergillus were detected in the ICU and were attributed to construction work that was undertaken near the hospital. All patients had pulmonary involvement, 3/8 were treated with L-Amphotericin B, 4/8 with a combination of L-Amphotericin B and Posaconazole and/or isavuconazole and 1/8 with isavuconazole. Four out of eight (50%) patients died [62]. Aspergillus fumigatus, Aspergillus niger and Aspergillus nidulans have been isolated [70], while mixed mold infections with Candida are also described in the literature [66,67]. Nidhya Ganesan et al. reported that among 60 biopsy samples from suspected rhino-maxillary/rhino-orbital mucormycosis post COVID-19, mucorales was isolated in 58 (96.67%) samples, aspergillus along with mucorales in 12 (20%) and a combination of mucorales and candida in 8 (13.33%) [24].

Neither 1,3-beta-D-glucan assay and galactomannan are positive in mucormycosis but can aid in the diagnosis of invasive pulmonary aspergillosis, which is recognized as a severe superinfection of COVID-19 pneumonia resulting in higher mortality. A positive serum or BAL galactomannan in a patient with compatible clinical presentation and imaging findings is indicative of invasive aspergillosis [127]. BAL galactomannan was measured in the study of R.H. Mehta et al. and was found positive (≥1) in 4/5 cases of COVID-19-associated pulmonary mucormycosis. In two cases, Aspergillus fumigatus was isolated in fungal culture, while in three cases, Aspergillus was identified in histopathological analysis [89]. Ultimately, mixed infection should be actively searched and isavuconazole is a potential empirical choice if mixed infection is suspected.

## 7. Imaging

CT and MRI imaging contribute to diagnosis and determine disease extension. Imaging evaluation in CAM is similar to mucormycosis not complicating COVID-19. Radiological findings include signs of sinusitis (thickened mucosa, opacification of the paranasal sinuses, air fluid levels), orbital invasion, cavernous sinus thrombosis and infiltration, internal carotid artery infiltration, cerebritis, cerebral infraction, thrombosis of the surface veins and dural venous sinuses, mycotic aneurysms, subarachnoid hemorrhage and abscess formation [8]. Bone destruction may be detected, as 79% of 96 patients with ROCM exhibited bony erosions [49]. The “black turbinate sign” on MRI is characterized by lack of contrast enhancement as a result of turbinate necrosis but can also be found in other circumstances [8]. CT and MRI-scan imaging are necessary for disease staging [35]. In pulmonary CAM, imaging findings recorded in CT were a reversed halo sign (8% of patients), consolidation (83%), cavitation (33%) and nodules (6%) [57]. Other CT findings include the halo sign, pleural effusion and wedge-shaped infiltrates [125].

## 8. Treatment

High-dose liposomal amphotericin B is the first line treatment for mucormycosis [6]. It is well established that antifungals are insufficient alone in treating ROCM and surgical debridement is needed, as the removal of necrotic tissues is necessary to allow antifungal penetration and is associated with better prognosis [128]. Lipid formulation of amphotericin B is preferred over Amphotericin B deoxycholate due to less nephrotoxicity and better CNS penetration. The starting dose of liposomal amphotericin B is in the range of 5–7.5 mg/kg/day and 10 mg/kg/day for brain involvement. Resistance of certain strains to amphotericin B is noted, with amphotericin B being ineffective against Cunninghamella bertholletiae and Apophysomyces elegans [124]. In India, due to shortages of liposomal amphotericin B, amphotericin B deoxycholate was also administered [83].

Isavuconazole and posaconazole are indicated as rescue therapy or in cases with preexisting renal failure [6]. However, due to deficiency of liposomal amphotericin B in India during the second wave of the COVID-19 pandemic, isavuconazole and posaconazole were also used as first line agents. Specifically, Soni et al. used oral posaconazole for mild cases of mucormycosis, along with surgical debridement [34]. With posaconazole DR tablets, increased bioavailability is achieved, while the drug can also be infused intravenously. Posaconazole suspension is not supported due to variable bioavailability, while steady state is achieved earlier with DR tablet formulation. Trough levels of posaconazole should be monitored and levels >1.0 μg/mL are pursued [129]. In a study by Atul Patel et al., 24.1% of patients (7/29) had a posaconazole trough level <1.2 μg/mL. They arbitrarily used 1.2 μg/mL as a cut off level, as posaconazole was used off label as a primary treatment for invasive mucormycosis due to shortage of amphotericin B. Antifungal therapy was changed to amphotericin B when subtherapeutic levels were detected [3]. Isavuconazole exhibits less hepatotoxicity and drug interactions, while therapeutic drug monitoring is not required and is not approved for prophylaxis. In immunocompromised patients or refractory cases, a combination of liposomal amphotericin B with echinocandins or posaconazole or isavuconazole may prove beneficial [124].

Other antifungals against mucormycosis were also tested during the pandemic and the recent surge of CAM due to unavailability of first-line agents. Specifically, Gupta et al. reported that susceptibility of Mucorales to itraconazole and terbinafine was species dependent, as 97.7% of R. oryzae and 36.5% of R. microsporus had MIC ≤ 2 µg/mL for itraconazole, while 85.2% of R. microsporus had MIC ≤ 2 µg/mL for terbinafine [126]. The use of iron chelators is under debate in the literature. Deferiprone and Deferasirox have in vitro activity against Mucorales [125], but have not been found to be beneficial for mucormycosis treatment [2]. No data were found in the literature concerning hyperbaric oxygen therapy. Patients with orbital involvement and intact vision can be managed with complementary transcutaneous retrobulbar amphotericin B (TRAMB) injections. A satisfactory response was noticed especially in patients without necrosis, as 40% of them showed improvement of visual acuity and/or ocular movement [31].

Early initiation of antifungal therapy and surgical removal of operable lesions are the mainstay of management for mucormycosis [39]. Surgical debridement of necrotic tissues is required among ROCM infections and multiple sessions may be needed to treat residual and recurrent disease [128].

In summary, management is based on three pillars: high-dose antifungal therapy, early source control when feasible and optimized management of associated conditions. Therefore, both strict glycemic control and restricted use of corticosteroids among COVID-19 patients requiring supplementary oxygen administration are essential to reverse predisposing factors [124].

## 9. Outcome

Median all-cause mortality rate was estimated to be 21.4% (Q1:14.3, Q3:46.2, IQR 31.9%). The median mortality among CAM patients with cerebral involvement was 50% (Q1:25, Q3:88.9, IQR 63.9%) and among CAM patients with pulmonary involvement, 80% (Q1:50, Q3:100, IQR 50%) (Table 2, Appendix A). When data were searched for the contribution of cerebral and pulmonary involvement to mortality rate, 142/237 (59.9%) patients who died (from 24 studies) were found with intracranial disease and 43/101 (42.6%) patients who died (from 13 studies) were found with pulmonary involvement (Appendix A). The median mortality rate in India was 18.2% (Q1:12.3, Q3:30.3, IQR 18), in Europe, 53.8%, and in the rest of the world, 39.8% (Q1:18.8, Q3:63.4, IQR 44.6).

Mortality rates vary in the literature. This is likely the result of the different forms of the disease, the challenging diagnosis, especially for pulmonary mucormycosis, and the association with either mild, moderate or severe COVID-19, which also affects mortality. In Europe, mortality ranges from 53.8% to 88% (Table 2). Data on mucormycosis-related mortality are lacking in the literature. In a national survey in Germany, including 13 patients with CAM from 6 tertiary care hospitals, all-cause mortality was 53.8% and mucormycosis-attributable mortality was 15.3% [76].

Delay in identification of CAM may be an important prognostic factor [128], as in studies where both mortality rates and the time period between COVID-19 and CAM diagnosis were recorded, median mortality was 33.7% (Q1:16.7, Q3:51, IQR 34.3%) when CAM diagnosis was performed after 15 days from COVID-19 diagnosis and 23.4% (Q1:15.5, Q3:50, IQR 34.5%) when it was performed ≤15 days post COVID-19 diagnosis.

In multivariate regression analysis in a study including 73 consecutive CAM patients, history of mechanical ventilation due to COVID-19 was associated with a 9-fold increased risk of death (*p* = 0.003) [45]. Other factors significantly associated with mortality were older age (>40 years), intracranial involvement, Hb1AC >9.1%, (n = 540) [31] advanced stage of ROCM, qSOFA ≥ 2 (n = 105) [58], chronic kidney disease, renal dysfunction during hospital stay, orbital involvement and tocilizumab use (n = 84) [54].

## 10. Limitations

The limitations of our review are the small number of prospective studies included (11/88) and the lack of studies comparing characteristics and outcomes of CAM patients with COVID-19 patients without mucormycosis. A high risk of bias was noted due to missing information on the incidence of CAM and heterogeneity in mortality rates was observed among studies mainly due to the different forms of mucormycosis included and diverse mortality endpoints used.

## 11. Conclusions

Our literature review suggests that COVID-19 may be complicated by secondary invasive fungal infections, including mucormycosis. Important geographical differences were identified and need to be taken into consideration. CAM was mainly reported in India, with an incidence of 0.27% to 3.36% among hospitalized COVID-19 patients. In India, near all reports were ROCM in patients with uncontrolled DM and history of corticosteroids intake. On the other hand, the most prevalent presentation in Europe was as pulmonary mucormycosis, particularly among hematologically immunocompromised patients with severe COVID-19. Patients with kidney transplant also seem to be exposed to a higher risk. Based on our findings, CAM was diagnosed a median of 17.4 days (IQR 7.5 days) post COVID-19. Since rapid diagnosis is crucial, molecular diagnostic techniques have to be generalized. Concomitant Aspergillus isolates were identified in 19/43 (44.2%) pulmonary mucormycosis reports and 89/863 (10.3%) CAM cases. Reported all-cause mortality was estimated to have a median of 21.4% (IQR 31.9%) (in ROCM with cerebral involvement 50% (IQR 63.9%), while in pulmonary, it was 80% (IQR 50%) and in India, it was 18.2% (IQR 18), while in Europe, it was 53.8%. When two weeks was used as the threshold for the diagnosis of CAM, median mortality was 23.4% (IQR 34.5%) vs. 33.7% (IQR 34.3%) after two weeks. Altogether, these studies revealed that an optimization of therapy is crucial, based on earlier high-dose antifungal therapy administration, early source control with repeated debridement when feasible, strict glycemic control and restriction of steroid therapy to COVID-19 patients with additional oxygen requirements.

## Figures and Tables

**Figure 1 diagnostics-12-03092-f001:**
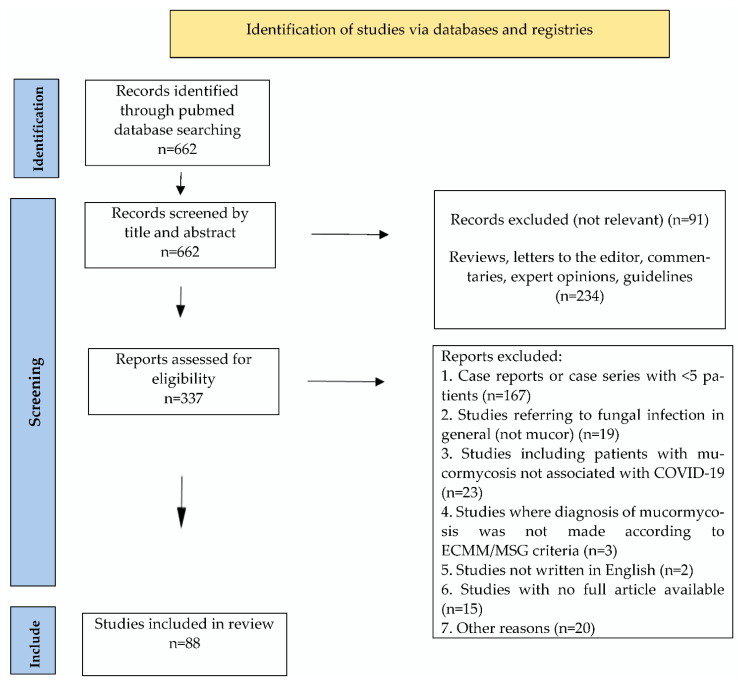
PRISMA Flowchart of published studies related to COVID-19 associated mucormycosis.

**Figure 2 diagnostics-12-03092-f002:**
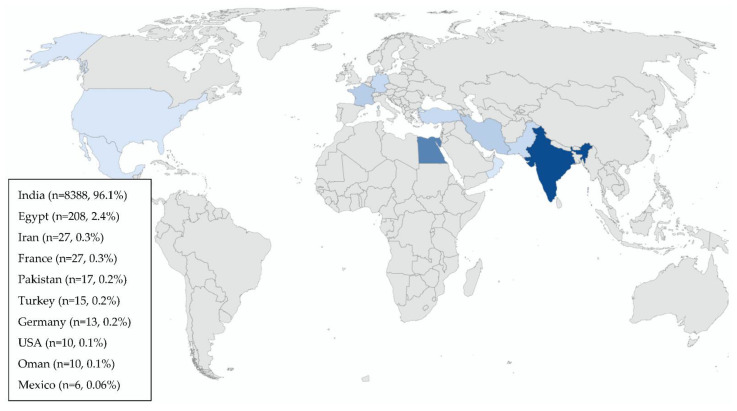
Geographical distribution of patients with COVID-19-associated mucormycosis (CAM) (n, %) reported in the 88 studies included.

**Figure 3 diagnostics-12-03092-f003:**
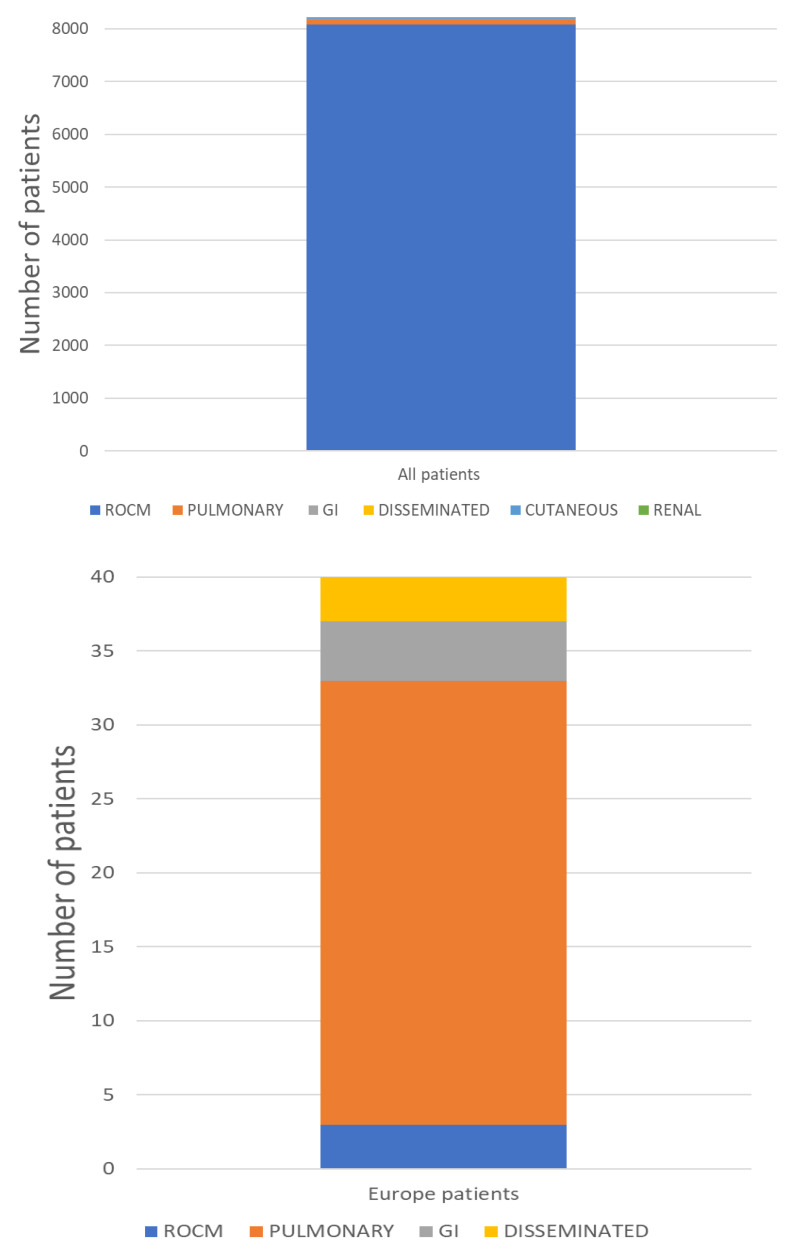
Clinical presentation of patients with CAM in total and in Europe.

**Table 1 diagnostics-12-03092-t001:** Studies’ characteristics.

Study	Study Design	Country	CAM Patients (n)	Setting/Departments	Quality Rating *
Said Ahmed WM et al. [29]	Case series	Egypt	14	Oral and Maxillofacial Surgery	Fair
Murthy R et al. [30]	Case series	India	111	NA	Poor
Walia S et al. [31]	Prospective cohort study	India	540	Eye Care centre	Good
Vare AA et al. [32]	Retrospective	India	67	Hospital	Good
Fouad YA et al. [33]	Multicentric, retrospective case series	Egypt	26	Ophthalmological center	Good
Soni K et al. [34]	Retrospective	India	145	Tertiary care center	Good
Metwally MI et al. [35]	Cross section study	Egypt	63	Radiology department	Fair
Arora U et al. [36]	Retrospective case-control study	India	152	Tertiary care center	Good
Jindal G et al. [37]	Prospective observational	India	15	Radiology department	Fair
Syed-Abdul S et al. [38]	Retrospective	India	214	Tertiary care center	Fair
Patel A et al. [3]	Retrospective	India	29	Tertiary care center	Good
Pruthi H et al. [39]	Case series	India	5	Referral center	Poor
Bansal SB et al. [40]	Retrospective	India	11	Tertiary care center	Good
Dulski TM et al. [41]	MMWR/CDC	USA	10	6 hospitals	Poor
Meshram HS et al. [42]	Retrospective cohort study	India	61	18 transplant centers	Good
Aggarwal SK et al. [43]	Case series	India	13	Referral center	Fair
Kulkarni R et al. [44]	Retrospective, multi-centre study	India	102	13 urban tertiary care centers	Good
Choksi T et al. [45]	Retrospective case–control study	India	73	Tertiary care center	Good
Kumar S et al. [46].	Prospective	India	287	Tertiary care center	Good
Mehta R et al. [47]	Case series	India	17	Head and Neck Surgery	Poor
Panwar P et al. [48]	Case series	India	7	Tertiary care center	Fair
Patel DD et al. [49]	Cross-sectional study	India	96	Radiology department	Fair
Vaid N et al. [50]	Observational	India	65	Tertiary care center	Poor
Goddanti N et al. [51]	Retrospective cohort study	India	300	ENT center	Good
Yadav T et al. [52]	Retrospective	India	50	Tertiary care center	Fair
Meshram VB et al. [53]	Retrospective	India	11	Nephrology and transplantation	Good
Zirpe K et al. [54]	Retrospective, observational	India	84	Tertiary care center	Fair
Alloush TK et al. [55]	Retrospective case series	Egypt	14	Tertiary care center	Fair
Pal P et al. [56]	Retrospective observational	India	10	NA	Poor
Danion F et al. [57]	Retrospective nationwide study	France	17	59 French mycology laboratories	Good
Nehara HR et al. [58]	Prospective observational	India	105	Tertiary care center	Fair
Pandiar D et al. [59]	Prospective observational	India	12	Outpatient center	Fair
Kumar S et al. [60]	Case–control study	India	55	Rural center	Good
Bilgic A et al. [61]	Retrospective cases series	India	38	6 centers	Good
Guemas E et al. [62]	Retrospective	France	10	ICU	Good
Kumar SG et al. [63]	Retrospective	India	101	Tertiary care centre	Fair
Mani S et al. [64]	Retrospective observational	India	89	Tertiary care center	Fair
Dravid A et al. [65]	Retrospective cohort study	India	59	Tertiary care center	Good
Naruka S et al. [66]	Case series	India	79	Tertiary care center	Good
Jain K et al. [67]	Prospective	India	95	Tertiary care center	Fair
Bhanuprasad K et al. [68]	Prospective	India	132	Hospital	Good
Desai EJ et al. [69]	Observational	India	100	ENT department	Poor
Nasir \n et al. [70]	Observational	Pakistan	10	NA	Fair
Gupta\s et al. [71]	Observational retrospective	India	56	ENT department	Fair
Joshi S et al. [4]	Retrospective	India	178	ENT department	Fair
Pradhan P et al. [72]	Retrospective	India	46	Otorhinolaryngology department	Poor
Mehta RNM et al. [73]	Prospective interventional study	India	215	NA	Good
Riad A et al. [74]	Case series	Egypt	7	4 hospitals	Fair
Guzmán-Castro S et al. [75]	Retrospective	Mexico	6	General hospital	Fair
Seidel D et al. [76]	Survey	Germany	13	6 tertiary care centers	Good
Gupta R et al. [77]	Multicentric observational	India	115	Tertiary care centers	Fair
Alfishawy M et al. [78]	Case series	Egypt	21	NA	Fair
Dave TV et al. [79]	Multi-centric retrospective	India	58	9 hospitals	Good
Selarka L et al. [80]	Prospective, observational, multi-centre	India	47	3 tertiary care centers	Fair
Avatef Fazeli M et al. [81]	Observational retrospective	Iran	12	Educational therapeutic hospital	Good
Mishra Y et al. [82]	Descriptive study	India	32	COVID-19 Care Centre	Good
Sen M et al. [83]	Retrospective, observational	India	2826	102 treatment centers	Fair
Pakdel F et al. [84]	Cross-sectional descriptive multicenter	Iran	15	5 COVID-19 centers	Good
Y M. Reddy et al. [85]	Case series	India	6	Department of Neurology	Fair
R.Arora et al. [86]	Cross-sectional study	India	60	Hospital	Fair
D.P Gupta et al. [87]	Prospective cross-sectional study	India	70	ENT department	Poor
M.Gautam et al. [88]	Prospective cohort study	India	30	Department of ophthalmology	Fair
R.M.Mehta et al. [89]	Case series	India	5	Department of pulmonology and Critical Care Medicine	Fair
Y.M.Reddy et al. [90]	Retrospective cohort study	India	31	Tertiary care hospital	Fair
S.P.Singh et al. [91]	Case series	India	6	Tertiary care center	Fair
M.Hada et al. [92]	Cross-sectional study	India	270	Tertiary care center	Fair
M. Kumar H et al. [93]	Case–control study	India	28	Tertiary care hospital	Good
S. Bhandari et al. [94]	Prospective study	India	235	Tertiary care center	Poor
M Chouhan et al. [95]	Ambispective interventional study	India	41	Tertiary care center	Fair
Y. Singh et al. [96]	Case series	India	13	Tertiary care center	Fair
S M Desai et al. [97]	Retrospective study	India	50	Radiology department	Good
A. Kumari et al. [98]	Retrospective study	India	20	Tertiary care center	Fair
S. Mitra et al. [99]	Case series	India	32	ENT department	Poor
A Ramaswami et al. [100]	Retrospective study	India	70	Emergency department	Fair
A.R. Joshi et al. [101]	Retrospective study	India	25	Radiology department	Poor
A. Patel et al. [102]	Retrospective observational study	India	187	16 healthcare centers	Good
S Sharma et al. [103]	Prospective observational study	India	23	Tertiary care centre	Fair
R. Kant et al. [104]	Case series	India	100	Tertiary care centre	Fair
C. Eker et al. [105]	Retrospective study	Turkey	15	Referral center for ENT care	Fair
A.K. Pandit et al. [106]	Case–control study	India	61	Tertiary care referral hospital	Good
S.F. Youssif et al. [107]	Retrospective cross-sectional	Egypt	33	Tertiary-care center	Good
A. Sekaran, et al. [108]	Retrospective study	India	30	Hospital	Fair
R. R. Shabana et al. [109]	Retrospective study	Egypt	30	Tertiary-care center	Fair
A. K Patel et al. [110]	Case–control study	India	64	Tertiary-care center	Good
H. D.D. Martins et al. [111]	Case series	Mexico, Brazil	6	Two referral services	Poor
S. Iqtadar et al. [112]	Case series	Pakistan	7	Hospital	Poor
A. Al Balushi et al. [113]	Case series	Oman	10	Secondary hospital	Fair
R. Soman et al. [114]	Case series	India	28	Hospital	Fair

ENT: Ear, Nose and Throat, ICU: Intensive Care Unit, MMWR: Morbidity and Mortality Weekly Report. * According to the Quality Assessment Tool for Case Series Studies by the National Heart, Lung and Blood Institute (NHLBI).

**Table 2 diagnostics-12-03092-t002:** Incidence of CAM among hospitalized COVID-19 patients, type of infection, invasive or non-invasive mechanical ventilation, risk factors and all-cause mortality.

Study	Incidence of CAM (%)	Type of Infection (%)	Cerebral Involvement /ROCM pts, n (%)	IMV or NIV n (%)	DM (% of CAM pts)	Steroids Intake (% of CAM pts)	All-Cause Mortality (%)
Said Ahmed WM et al. [29]	NA	Maxillary osteomyelitis	0/14 (0%)	NA	64.2% DM 35.7% with temporary post-COVID-19 hyperglycemia	NA	NA
Murthy R et al. [30]	NA	RO	0/111 (0%)	NA	NA	NA	NA
Walia S et al. [31]	NA	SN (100%), O (51.85%), C (9.44%), Cu (1.85%), P (0.18%).	51/529 (9.6%)	NA	97.96%	84.85%	9.25%
Vare AA et al. [32]	1.36%	ROCM	3/67 (5%)	18/67 (27%)	90%	84%	34%
Fouad YA et al. [33]	NA	O	0/26 (0%)	NA	96.2%	76.9%	46,2%
Soni K et al. [34]	NA	ROCM	29/145 (20%)	NA	86.2%	65%	18%
Metwally MI et al. [35]	NA	Head and neck	8/63 (12.7%)	NA	80.9%	82.5%	17.5%
Arora U et al. [36]	NA	RS (29%), RO (47.3%), ROCM (14.5%), O (1.3%), RO/palatal (5.3%), Cu (0.6%), P (1.3%), D (0.6%)	22/148 (14.9%)	NA	92.1%	65.8%	NA
Jindal G et al. [37]	NA	ROCM	9/15 (60%)	NA	100%	80%	6.6%
Syed-Abdul S et al. [38]	NA	NA	NA	NA	NA	NA	NA
Patel A et al. [3]	NA	RO (96.5%), P (3.4%)	0/28 (0%)	NA	NA	NA	NA
Pruthi H et al. [39]	NA	P	NA	0/5 (0%)	100%	NA	80%
Bansal SB et al. [40]	10.8%	RO (91%), P (9%)	NA	0/11 (0%)	64%, 36% developed transient hyperglycemia	100%	18.2%
Dulski TM et al. [41]	NA	RO (10%), ROCM (30%), P (30%), D (20%), GI (10%)	3/4 (75%)	5/10 (50%)	80%	90%	60%
Meshram HS et al. [42]	4.4%	ROCM (91.8%), P (8.2%)	11/42 (26.2%)	0/61 (0%)	24.6%	44%	26.2%
Aggarwal SK et al. [43]	NA	ROCM	4/13 (30.8%)	NA	92.3%	92.3%	15.4%
Kulkarni R et al. [44]	2.1% (1 centre)	ROCM	12/102 (11.8%)	NA	81.6%	NA	51%.
Choksi T et al. [45]	NA	ROCM	6/73 (2%)	17/73 (23.3%)	74%	98%	36%
Kumar S et al. [46]	NA	ROCM	60/287 (21%)	NA	80%	NA	NA
Mehta R et al. [47]	NA	ROCM	0/17 (0%)	NA	100%	NA	NA
Panwar P et al. [48]	NA	ROCM	0/7 (0%)	NA	100%	42.8%	0%
Patel DD et al. [49]	NA	ROCM	21/96 (21.9%)	6/96 (6.3%)	71.8%	82.3%	NA
Vaid N et al. [50]	NA	ROCM	NA	NA	33.8%	100%	10.7%
Goddanti N et al. [51]	NA	ROCM	NA	NA	95.7%	79%	NA
Yadav T et al. [52]	NA	ROCM	25/50 (50%)	NA	86%	44%	NA
Meshram VB et al. [53]	NA	ROCM (90.9%), P (9%)	3/10 (30%)	0/11 (0%)	54.5%	100%	27%
Zirpe K et al. [54]	NA	ROCM	20/84 (23.8%)	NA	64.3%	83.3%	15.5%
Alloush TK et al. [55]	NA	ROCM	9/14 (64.2%)	0/14 (0%)	92.8%	NA	21.4%
Pal P et al. [56]	NA	ROCM	3/10 (30%)	ΝA	70%	80%	30%
Danion F et al. [57]	NA	P(53%), GI (18%), ROCM (12%), D (18%)	NA	13/17 (76.5%)	47%	76.5%	88%
Nehara HR et al. [58]	NA	ROCM	18/105 (17.1%)	NA	78.1%	66.3%	19.05%
Pandiar D et al. [59]	NA	Oral	0/12 (0%)	NA	66,7%	NA	NA
Kumar S et al. [60]	NA	NA	NA	0/55 (0%)	83.6%	98.2%	16%
Bilgic A et al. [61]	2.5%	ROCM	NA	6/38 (16%)	50%	100%	5%
Guemas E et al. [62]	7.1%	P	NA	NA	20%	90%	50%
Kumar SG et al. [63]	NA	ROCM	44/101 (43.6%)	NA	94%	80.1%	17.8%
Mani S et al. [64]	NA	ROCM	4/89 (4.5%)	NA	96%	92%	3.4%
Dravid A et al. [65]	NA	ROCM (98.3%), D (1.7%)	26/58 (44.8%)	3/59 (5.1%)	89.8%	100%	25.4%
Naruka S et al. [66]	NA	ROCM	9/79 (11.4%)	NA	100%	NA	18.18%
Jain K et al. [67]	NA	ROCM	3/95 (3.2%)	NA	77%	100%	5.2%
Bhanuprasad K et al. [68]	NA	ROCM	39 (29.5%)	3/132 (2.3%)	97.7%	55.3%	9.8%
Desai EJ et al. [69]	NA	ROCM	0/100 (0%)	NA	80%	NA	20%
Nasir\n et al. [70]	0.35%	P (60%), ROCM (40%)	4/4 (100%)	3/10 (30%)	70%	80%	70%
Gupta \s et al. [71]	NA	ROCM	4/56 (7.1%)	NA	85%	66%	16%
Joshi S et al. [4]	NA	ROCM	22/178 (12.4%)	5/178 (2.8%)	74.2%	52.8%	15%
Pradhan P et al. [72]	NA	ROCM	10/46 (21.7%)	NA	95.65%	89.1%	19.5%
Mehta RNM et al. [73]	NA	ROCM	33/215 (15.3%)	NA	91%	88%	12.1%
Riad A et al. [74]	NA	ROCM	7/7 (100%)	NA	85.7%	100%	0%
Guzmán-Castro S et al. [75]	0.04%	ROCM (83.3%), P(16.6%)	5/5 (100%)	2/6 (33.3%)	83.3%	100%	83.3%
Seidel D et al. [76]	2 centres: 0.67%, 0.58% ICU: 1.47%, 1.78%	P(84.6%), ROCM (7.7%), GI (7.7%)	1/1 (100%)	11/13 (84.6%)	23.07%	84.6%	53.8%
Gupta R et al. [77]	NA	ROCM	25/115 (21.7%)	13/115 (11.3%)	85.2%	100%	21.7%
Alfishawy M et al. [78]	NA	ROCM (95.2%), P (4.8%)	5/20 (25%)	NA	90%	100%	33.3%
Dave TV et al. [79]	NA	ROCM	19/58 (33%)	NA	74%	NA	34%
Selarka L et al. [80]	1.8%	ROCM	9/47 (19.1%)	20/47 (42.6%)	76.6%	100%	23.4%
Avatef Fazeli M et al. [81]	NA	ROCM	0/12 (0%)	1/12 (8.3%)	83.33%	75%	66.7%
Mishra Y et al. [82]	3.36%	ROCM	0/32 (0%)	NA	87.5%	93%	12.5%
Sen M et al. [83]	NA	ROCM	539/2826 (19.1%)	114/1602 (7.1)	78%	87%	14%
Pakdel F et al. [84]	NA	ROCM	7/15 (46%)	1/15 (6.7%)	86%	46.6%	47%
Y M. Reddy et al. [85]	NA	RO	0/6 (0%)	NA	100%	66.7%	100%
R. Arora et al. [86]	NA	ROCM	6/60 (10%)	NA	98.3%	63.3%	NA
D.P Gupta et al. [87]	NA	ROCM	NA	NA	100%	NA	5.7%
M.Gautam et al. [88]	NA	ROCM	NA	NA	100%	66.7%	0%
R.M.Mehta et al. [89]	NA	P	NA	4/5 (80%)	80%	100%	80%
Y.M.Reddy et al. [90]	NA	ROCM	NA	NA	100%	80.6%	35.5%
S.P.Singh et al. [91]	NA	RO	0/6 (0%)	0/6 (0%)	100%	66.7%	16.7%
M.Hada et al. [92]	NA	ROCM	54/270 (20%)	NA	92.2%	72%	NA
M. Kumar H et al. [93]	NA	ROCM (85.7%), P (14.3%)	15/24 (62.5%)	6/28 (21.4%)	75%	70.4%	73.9%
S. Bhandari et al. [94]	NA	NA	NA	NA	86.8%	84.3%	NA
M Chouhan et al. [95]	NA	ROCM	9/41 (21.9%)	NA	97.6%	87.8%	9.8%
Y. Singh et al. [96]	NA	ROCM (92.3%), P (7.7%)	2/12 (16.7%)	10/13 (76.9%)	61.5%	84.6%	69.2%
S M Desai et al. [97]	NA	ROCM	3/50 (6%)	NA	82%	84%	30%
A. Kumari et al. [98]	NA	ROCM	4/20 (20%)	NA	80%	80%	30%
S. Mitra et al. [99]	NA	ROCM	NA	NA	100%	78.1%	NA
A Ramaswami et al. [100]	NA	ROCM	17/70 (24.3%)	NA	70%	70%	NA
A.R. Joshi et al. [101]	NA	ROCM	7/25 (28%)	12/25 (48%)	88%	100%	56%
A. Patel et al. [102]	7 centers: 0.27% (general wards)	ROCM (86.1%), P (8.6%), renal (0.5%), other (e.g., Cu, GI) (2.7%), D (2.1%)	44/161 (27.3%)	NA	60.4%	78.1%	44.1%
S Sharma et al. [103]	NA	ROCM	2/23 (8.7%)	NA	91.3%	100%	NA
R. Kant et al. [104]	NA	ROCM (96%), P (4%)	11/96 (11.5%)	NA	95%	81%	13%
C. Eker et al. [105]	NA	ROCM	9/15 (60%)	NA	100%	NA	33.3%
A.K. Pandit et al. [106]	NA	ROCM	30/56 (53.6%)	NA	85.7%	53.6%	30.6%
S.F. Youssif et al. [107]	7.6%	ROCM	32/33 (97%)	NA	63.6%	NA	90.9%
A. Sekaran, et al. [108]	NA	ROCM	6/30 (20%)	8/30 (26.7%)	100%	90%	16.7%
R. R. Shabana et al. [109]	NA	ROCM	4/30 (13.3%)	1/30 (3.3%)	90%	66.6%	20%
A. K Patel et al. [110]	NA	ROCM (92.2%), P (7.8%)	5/59 (8.5%)	NA	75%	90.6%	4.7%
H. D.D. Martins et al. [111]	NA	ROCM	0/6 (0%)	NA	83.3%	NA	16.7%
S. Iqtadar et al. [112]	NA	ROCM	NA	0/7 (0%)	71.4%	100%	14.3%
A. Al Balushi et al. [113]	NA	ROCM	3/10 (30%)	6/10 (60%)	100%	30%	60%
R. Soman et al. [114]	NA	ROCM (78.6%), P (21.4%)	5/22 (22.7%)	NA	NA	NA	25%

C: Cerebral, Cu: Cutaneous, DM: Diabetes Mellitus, D: Disseminated, GI: gastrointestinal, IMV: Invasive Mechanical Ventilation, NA: Not Available, NIV: Non-invasive ventilation, O: Orbital, P: Pulmonary, RO: Rhino Orbital, ROCM: Rhino-Orbital-Cerebral Mucormycosis, RS: Rhino-sinus, SN: Sinonasal.

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
