# Peer review of "Clinical Phenotypes of COVID-19 Associated Mucormycosis (CAM): A Comprehensive Review"

_diagnostics, 2022, doi:10.3390/diagnostics12123092_

Round 1
Reviewer 1 Report
The article presents a systematic review of mucormycosis associated with COVID-19. The authors compiled and described studies about the incidence of CAM, risk factors, clinical manifestation, diagnosis, and treatment. This is a relevant review in the field of clinical mycology.
There are some minor points and corrections in the text that should be addressed.
Lines 39-40: the number of cases seems too high for the period of three months, is this correct? is the reference number 3 correct?
The pathophysiology topic (number 2) seems to be out of place in the general outline of the article. I recommend a different structure to the article. It does not seem to me that should be a results section, given its review nature. Although it is a systematic review, I think that the results are in fact a description of the findings in the articles included. It would be clearer to the readers. I also suggest a discussion and critical review of the studies included.
Lines 115-116: incidence of CAM in which period?
Lines 139, 159, 171: tables 2 should be table 2.
Line 168: the phrase should be rewritten. Suggestion: standard therapy for severe COVID-19 which increases the risk of secondary infections.
Line 469: the phrase “our findings” should be rewritten. This is not an original article, so it should be clear that the authors do not have any results, but a thorough review of studies about CAM. I believe it is more suitable to say “Altogether, these studies revealed that a optimization of therapy is crucial, based on….”.
Figure 1 needs to be formatted to exclude copy-paste mistakes. The legend should be changed to make it clear that the studies included are related to CAM or mucormycosis.
The figures are very confusing. The graphics should be reformatted, and the legends should have a better description of the data.
Figures 2, 3 and 4: there are 2 graphics in these figures, I recommend naming them A and B and to describe in the legend what they mean. It is not clear in the text nor the figure what the graphics represent.
Reviewer 2 Report
Almyroudi M.P. and colleagues performed a narrative review about COVID-19 associated mucormycosis.
They report data about more than 8000 patients, exploring in a very exhaustive way all the issues related to diagnosis and management of the disease.
A lot of reviews about secondary fungal infections in COVID-19 are available, and almost everything has been said about CAM. However, I think this study summarize the characteristics of the greatest number of CAM cases to date.
This review is too long and, sometimes, difficult to read because of the amount of data reported. I would try to shorten the text as much as possible. Authors could maintain the actual partition of the results (incidence, risk factors etc…) but each section should be squeezed.
The introduction needs more references.
The main core of this review are clinical data, so I would delete section 2 about pathophysiology. The Authors could briefly summarize the main issues about pathophysiology in the introduction.
Figure 2 could be a map, instead of a cake graph. The histogram is not really useful.
Figure 3 should be improved (columns and legend overlap)
Figure 4 is not useful.
Table S1 data are already included in Table 2, so delete it.
Figure 1: there is probably an issue with the screenshot of this figure, because editing tools are clearly visible on the “screening” panel.
Line 168-169: please report a reference to support this statement.
Line 104-107. Why have you calculated median mortality? I think it’s not a meaningful data. I would simply summarize mortality data of the studies as a range (minimum and maximum value) or you could calculate a cumulative mortality (total deaths reported/total cases reported*100), even if this approach has obvious limits. This is not a systematic review, a metanalysis nor a pooled analysis, so it is not expected to report a precise estimate of CAM mortality. Just report and summarize the data from the literature.
Line 430-434: please report the reference.
When you report IQR, please state Q1 and Q3 separately.
In conclusion, this paper would be suitable for publication after revisions.
Moderate English editing is required before publication.
Round 2
Reviewer 2 Report
I would like to thank you the Authors for adressing most of my suggestions.